# Velocity Filtering Using Quantum 3D FFT

**Georgia Koukiou *** and **Vassilis Anastassopoulos ***

Electronics Laboratory, Physics Department, University of Patras, 26504 Patras, Greece
* Correspondence: gkoukiou@upatras.gr (G.K.); vassilis@upatras.gr (V.A.); Tel.: +30-26-1099-6147 (G.K. & V.A.)

**Abstract:** In this work, the quantum version of 3D FFT is proposed for constructing velocity filters. Velocity filters are desirable when we need to separate moving objects with a specific velocity range in amplitude and direction in a rapidly changing background. These filters are useful in many application fields, such as for monitoring regions for security reasons or inspecting processes in experimental physics. A faster and more attractive way to implement this filtering procedure is through 3D FFT instead of using 3D FIR filters. Additionally, 3D FFT provides the capability to create banks of ready-made filters with various characteristics. Thus, 3D filtering is carried out in the frequency domain by rejecting appropriate frequency bands according to the spectral content of the trajectory of the object to be isolated. The 3D FFT procedure and the corresponding inverse one are required in the beginning and end of the filtering process. Although 3D FFT is computationally effective, it becomes time-consuming when we need to process large data cubes. The implementation of velocity filters by means of the quantum version of 3D FFT is investigated in this work. All necessary quantum circuits and quantum procedures needed are presented in detail. This proposed quantum structure results in velocity filtering with a short execution time. For this purpose, a review of the necessary quantum computational units is presented for the implementation of quantum 3D FFT and representative examples of applications of velocity filtering are provided.

**Keywords:** quantum Fourier transform; quantum circuits; velocity filters; filter banks

## 1. Introduction

Fast Fourier transform (FFT) provides a means for frequency analysis and filtering, avoiding the operation of convolution. The quantum version of Fourier transform (QFT) provides an opportunity to quickly obtain the results derived by FFT. The use of QFT has been investigated for various applications. Accordingly, in [1], it is proved that a quantum version of the filtering operation can be achieved, even though the quantum convolution of two sequences is physically impossible. There are important differences between classical and quantum implementations for image filtering. These differences are analyzed in [1], and it is shown that the major advantage of the quantum approach lies in the exploitation of the efficient implementation of QFT. A common approach to image filtering is to convolve the image with a filter function, which in the frequency domain translates into a multiplication operation. However, there are classical processing operations that cannot be directly applied to quantum images, such as convolution and correlation [2]. In [3], a survey is provided with some topics on the properties of quantum gates and their assembly into interesting quantum circuits. The role of reversibility in the theory of computation and an early discussion of gate and circuit constructions in reversible computation are also provided.

The application of QFT within the field of quantum computation has been extensively presented in [4]. Shor's algorithm, phase estimation, and computing discrete logarithms are some classic examples of its use. These special properties of quantum algorithms have resulted in novel solutions to problems difficult to be solved on a classical computer. QFT has been used in several applications [5–7]. Since QFT is the core to a lot of quantum algorithms, current research mainly focuses on its effective realization [7–13]. These studies

discuss quantum information issues [7], approximate QFT and decoherence [8], Shor's algorithm [9,10], QFT for phase estimation [11], and quantum circuitry [12,13]. Quantum edge detection is carried out in [14] based on double-chain quantum genetic algorithms. Hybrid quantum-classical networks for image generation are proposed in [15].

A review of quantum image processing is presented in [16], revealing the possibilities for intensive image-processing procedures due to the powerful parallel computing capabilities of quantum computers. In [17], a quantum implementation of the FFT algorithm composed of a combination of quantum gates is proposed. QFT is implemented in [18] on a 3-qubit nuclear magnetic resonance (NMR) quantum computer to extract the periodicity of an input state. A fast quantum image component-labeling algorithm is proposed in [19], which is the quantum counterpart of the classical local operator technique. A quantum color image encryption algorithm is designed in [20] based on geometric transformation and intensity channel diffusion. A framework of quantum image filtering in the spatial domain is proposed in [21]. A quantum image median filtering approach is proposed and its corresponding quantum circuit is designed in [22]. The main idea of the approach is that first, the classical image is converted into a quantum version based on the novel enhanced quantum representation (NEQR) of digital images, and then, a unique quantum module is designed to realize the median calculation of neighborhood pixels for each pixel point in the image. In [23], the authors consider QFT-based color-image-filtering operations and their applications in image smoothing, sharpening, and selective filtering using quantum frequency domain filters. The proposed quantum filters use the principle of quantum oracles to implement the filter function.

The 3D FFT technique is useful in numerous physical problems. Four of these problems are listed next in order to provide the reader with a sense of the potential applicability of QFT. First-principles methods based on density functional theory (DFT) where the wave functions are expanded in plane waves (Fourier components) are the most widely used approaches for electronic structure calculations in materials science [24]. The scaling of this method depends critically on having an efficient parallel 3D FFT that minimizes communications and calculations. First-principles methods based on DFT in the Kohn–Sham (KS) [25] formalism are the most widely used approaches for electronic structure calculations in materials science. The most common implementation of this approach involves the expansion of the wave functions in plane waves (Fourier components) and the use of pseudopotentials to replace the nucleus and core electrons. In this implementation, parallel 3D FFT is required to transform the electronic wave functions from Fourier space to real space in order to construct the charge density. The 3D Fourier forward modeling of 3D density sources is capable of providing 3D gravity anomalies coinciding with the meshed density distribution within the whole source region [26]. Forward modeling of potential fields' anomalies is essential for geophysical interpretation and inversion. An implicit split-operator FFT algorithm for the numerical solution of the time-dependent Schrodinger equation is implemented for the electronic structure of $H_2^+$ and $H_2$ in [27]. In this article, an algorithm appropriate for 3D applications is implemented that is implicit and thus overcomes the difficulty of the non-conservation of energy.

The tracking and isolation of moving objects with a specific range of speed are a challenging research topic in the field of automotive application [28–31]. To cope with this issue, velocity filters have been used in the past for localizing and monitoring moving objects in image sequences or otherwise 3D imagery [32–34]. Filter banks are used for fast implementation of the localization and monitoring of moving vehicles. These banks are built using 3D FFT to perform directional filtering [28–35].

In this work, the way that basic quantum circuits are combined to build up the QFT structure is extensively presented. This structure is applied in the well-known separable procedure in order to implement 3D QFT. This way, the 3D spectral content of the data cube is evaluated. A quantum oracle is used for isolating the necessary frequency components in the 3D spectral cube that correspond to the required trajectory. In fact, a variety of quantum oracles are represented by the quantum filter used in order to realize the necessary filter

bank. Inverse QFT (IQFT) is applied at the output of the quantum oracle to obtain the final result with the isolated moving object (trajectory). The implementation of velocity filters by means of the quantum version of 3D FFT results in fast filtering procedures, which are necessary for discriminating objects' velocities. QFT can calculate the Fourier transform of a vector of size N with time complexity $\mathcal{O}(log_2^2 N)$ compared to the classical complexity of $\mathcal{O}(N log_2 N)$ [36]. However, if one wants to measure the full output state, then the QFT complexity becomes $\mathcal{O}(N log_2^2 N)$, thus losing its apparent advantage, indicating that the advantage is fully exploited for algorithms when only a limited number of samples is required from the output vector, as is the case in many quantum algorithms. Accordingly, for a signal of 1024 samples, QFT requires operations of the order of 100, while classical FFT requires operations of the order of 10,000.

The paper is organized in the following way. In Section 2, the quantum theory for supporting QFT is presented. The velocity filtering approach is provided in Section 3. In Section 4, the use of QFT for implementing velocity filters is analyzed. Experimental results are presented in Section 5. Finally, conclusions are drawn in Section 6.

## 2. Quantum Theory

### 2.1. Quantum Fourier Transform Theory

Fourier transform occurs in many different versions in all areas from signal processing to complexity theory to data compression [37–39]. QFT is the classical discrete Fourier transform applied to a vector of amplitudes of a quantum state, where we usually consider vectors of length $N$.

Discrete Fourier transform acts on a vector $(x_0, x_1, \ldots, x_{N-1}) \in C^N$ and maps it to a vector $(y_0, y_1, \ldots, y_{N-1}) \in C^N$ according to the formula

$$y_k = \frac{1}{\sqrt{N}} \sum_{j=0}^{N-1} x_j \omega_N^{jk}, \quad k = 0, 1, 2, 3, \cdots, N-1 \tag{1}$$

where $\omega_N^{jk} = e^{\frac{2\pi i}{N} jk}$ and $\omega_N^{j}$ is the *j*-th root of unity.

Similarly, QFT acts on a quantum state $|x\rangle = \sum_{j=0}^{N-1} x_j |j\rangle$ and maps it to a quantum state $|y\rangle = \sum_{k=0}^{N-1} y_k |k\rangle$ according to the formula

$$y_k = \frac{1}{\sqrt{N}} \sum_{j=0}^{N-1} x_j \omega_N^{jk}, \quad k = 0, 1, 2, 3, \cdots, N-1 \tag{2}$$

Since $\omega_N^{jk}$ is a rotation, IQFT acts similarly:

$$x_j = \frac{1}{\sqrt{N}} \sum_{k=0}^{N-1} y_k \omega_N^{-jk}, \quad j = 0, 1, 2, 3, \cdots, N-1 \tag{3}$$

In case $|j\rangle$ is a basis state, QFT can also be expressed as the map

$$|j\rangle \rightarrow \frac{1}{\sqrt{N}} \sum_{k=0}^{N-1} \omega_N^{jk} |k\rangle \tag{4}$$

Equivalently, QFT can be viewed as a unitary matrix (or quantum gate) acting on quantum-state vectors, where the unitary matrix $F_N$ is given by

$$F_N = \frac{1}{\sqrt{N}} \begin{bmatrix} 1 & 1 & 1 & 1 & & 1 \\ 1 & \omega & \omega^2 & \omega^3 & & \omega^{N-1} \\ 1 & \omega^2 & \omega^4 & \omega^6 & & \omega^{2(N-1)} \\ 1 & \omega^3 & \omega^6 & \omega^9 & \cdots & \omega^{3(N-1)} \\ \vdots & \vdots & \vdots & \vdots & & \vdots \\ 1 & \omega^{N-1} & \omega^{2(N-1)} & \omega^{3(N-1)} & & \omega^{(N-1)(N-1)} \end{bmatrix} \tag{5}$$

where $\omega = \omega_N$.

Most of the properties of QFT follow from the fact that it is a unitary transformation. This can be checked by performing matrix multiplication and ensuring that the relation $FF^+ = F^+F = I$ holds, where $F^+$ is the Hermitian adjoint of $F$. Alternately, one can check that orthogonal vectors of norm 1 get mapped to orthogonal vectors of norm 1.

From the unitary property, it follows that the inverse of quantum Fourier transform is the Hermitian adjoint of the Fourier matrix, so $F^{-1} = F^+$. Since there is an efficient quantum circuit implementing QFT, the circuit can be run in reverse to perform IQFT. Thus, both transforms can be efficiently performed on a quantum computer.

QFT transforms between two bases, the Fourier basis and the computational basis (Z). The H gate is single-qubit QFT, and it transforms between the Z-basis states $|0\rangle$ and $|1\rangle$ to the X-basis states $|+\rangle$ and $|-\rangle$. In the same way, all multi-qubit states in the computational basis have corresponding states in the Fourier basis. QFT is simply the transform that transforms between these bases.

### 2.2. The Circuit Implementation of QFT

The circuit implementation of QFT makes use of two gates. One of them is the single-qubit Hadamard gate $H = \frac{1}{\sqrt{2}}\begin{pmatrix} 1 & 1 \\ 1 & -1 \end{pmatrix}$, and the other is the phase gate $R_m = \begin{pmatrix} 1 & 0 \\ 0 & e^{2\pi i/2^m} \end{pmatrix}$.

Consider how QFT operates on a single-qubit state (1-qubit QFT) $|\psi\rangle = a|0\rangle + b|1\rangle$, where $x_0 = a$, $x_1 = b$ and $N = 2$.

$$y_0 = \frac{1}{\sqrt{2}}\left( a \cdot e^{\frac{2\pi i}{2} 0 \times 0} + b \cdot e^{\frac{2\pi i}{2} 1 \times 0} \right) = \frac{1}{\sqrt{2}}(a + b) \tag{6}$$

and

$$y_1 = \frac{1}{\sqrt{2}}\left( a \cdot e^{\frac{2\pi i}{2} 0 \times 1} + b \cdot e^{\frac{2\pi i}{2} 1 \times 1} \right) = \frac{1}{\sqrt{2}}(a - b) \tag{7}$$

so the final result is the state

$$U_{QFT}|\psi\rangle = \frac{1}{\sqrt{2}}(a + b)|0\rangle + \frac{1}{\sqrt{2}}(a - b)|1\rangle \tag{8}$$

This operation is exactly the result of applying the Hadamard gate to the qubit. If we apply the Hadamard gate to the state $|\psi\rangle = 0 + b|1\rangle$, we obtain a new state:

$$\frac{1}{\sqrt{2}}(a + b)|0\rangle + \frac{1}{\sqrt{2}}(a - b)|1\rangle \equiv \bar{a}|0\rangle + \bar{b}|1\rangle \tag{9}$$

The Hadamard gate H for n qubits is given as

$$H_{2^n} = H_2 \otimes H_{2^{n-1}}, \; 2 \leq n \in N \tag{10}$$

where $\otimes$ is the Kronecker product, i.e., for $n = 2$, we have the Hadamard gate $H_4$

$$H_{2^2} = H_4 = H_2 \otimes H_{2^{2-1}} = H_2 \otimes H_2 = \frac{1}{\sqrt{2}}\begin{pmatrix} 1 & 1 \\ 1 & -1 \end{pmatrix} \otimes \frac{1}{\sqrt{2}}\begin{pmatrix} 1 & 1 \\ 1 & -1 \end{pmatrix} = \frac{1}{2}\begin{pmatrix} 1 & +1 & 1 & 1 \\ 1 & -1 & 1 & -1 \\ 1 & 1 & -1 & -1 \\ 1 & -1 & -1 & 1 \end{pmatrix} \tag{11}$$

Given these two gates, a circuit implementation of n-qubit QFT is shown in Figure 1 [34,35,37].

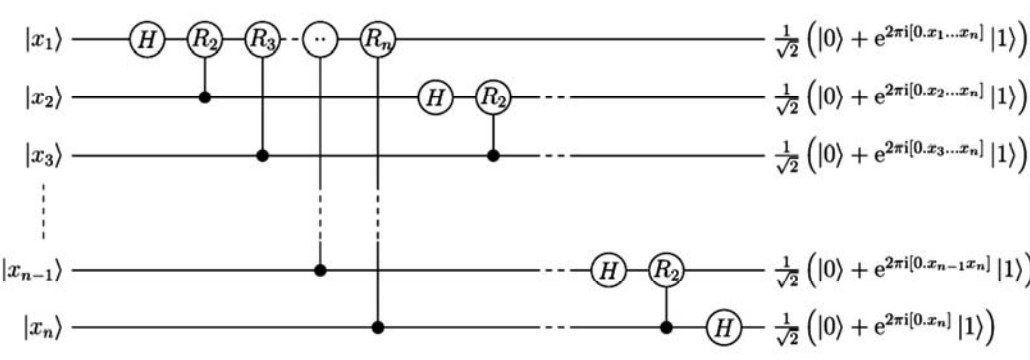

**Figure 1.** The circuit implementation of n-qubit QFT using the Hadamard gate $H$ and the phase gate $R_m$.

The basis states that enumerate all possible states of the n qubits are

$$|x\rangle = |x_1\rangle \otimes |x_2\rangle \otimes |x_3\rangle \otimes \cdots \otimes |x_{n-1}\rangle \otimes |x_n\rangle \tag{12}$$

where $|x_k\rangle$ indicates that qubit $k$ is the state $x_k$, with $x_k$ being either 1 or 0. The basis state index $x$ is the binary number encoded by $x_k$, with $x_1$ being the most significant bit. So, we can write the QFT as

$$QFT(|x\rangle) = \frac{1}{\sqrt{N}} \otimes_{k=1}^{n} \left(|0\rangle + e^{2\pi i x/2^k}|1\rangle\right) \tag{13}$$

After rearranging the sum and the products and expanding $\sum_{y=0}^{N-1} = \sum_{y_1=0}^{1} \sum_{y_2=0}^{1} \cdots \sum_{y_n=0}^{1}$, the action of the QFT can be expressed by

$$QFT\left(|x_1 x_2 x_3 \cdots x_{n-1} x_n\rangle\right) = \frac{1}{\sqrt{N}}\left(|0\rangle + e^{\frac{2\pi i x_n}{2}}|1\rangle\right) \otimes \left(|0\rangle + e^{\frac{2\pi i x_{n-1}}{2} + \frac{2\pi i x_n}{2^2}}|1\rangle\right) \otimes \cdots \otimes \left(|0\rangle + e^{\frac{2\pi i x_1}{2} + \frac{2\pi i x_2}{2^2} + \frac{2\pi i x_3}{2^3} + \cdots + \frac{2\pi i x_{n-1}}{2^{n-1}} + \frac{2\pi i x_n}{2^n}}|1\rangle\right) \tag{14}$$

i.e., for 3-qubit QFT

$$|y\rangle = \frac{1}{\sqrt{2^3}}\left[\left(|0\rangle + e^{\frac{2\pi i x_3}{2}}|1\rangle\right) \otimes \left(|0\rangle + e^{\frac{2\pi i x_2}{2} + \frac{2\pi i x_3}{2^2}}|1\rangle\right) \otimes \left(|0\rangle + e^{\frac{2\pi i x_1}{2} + \frac{2\pi i x_2}{2^2} + \frac{2\pi i x_3}{2^3}}|1\rangle\right)\right] \tag{15}$$

In a similar way, one can extend to N-qubit QFT if a larger amount of information is to be processed.

## 3. Theory of Velocity Filtering

Studies [28,29] have carried out velocity filtering using filter banks created using conventional FFT, and simultaneously, they are selected by means of the specific spectral content required. In this work, all necessary spectral quantities are evaluated using QFT for radically accelerating the mathematical calculations. A bank of velocity filters [28] is necessary for separating multiple objects with different velocities in a sequence of frames. In [28], the 3D FFT transformation of a large number of objects with a variety of different velocities is applied. The proposed approach is based on experimentation and avoids using theoretical concepts. Accordingly, in this study, an object moving each time with a different velocity and in various directions was used in order to construct a filter bank. Multiple moving objects can be isolated from other objects with different velocities or from objects with the same amplitude of velocity but having different directions. The simulated datasets that were used in order to create the spectral signatures of different moving objects consisted of 256 frames of 256 × 256 pixels each. Accordingly, a data cube

(shown in Figure 2) was formed of $256^3$ pixels. The number $256 = 2^8$ was selected to fit the FFT requirements for fast evaluation of the 3D spectrum.

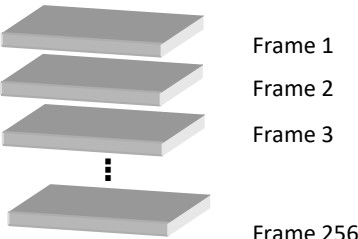

**Figure 2.** The 256 frames of $256 \times 256$ pixels each (data cube).

The time parameter is considered the distance from frame to frame. Based on this, the amplitude of the radial velocity of each object is referred to as the number of pixels it comes across from one frame to the next. A simple example of one object of size $10 \times 10$ pixels that is moving with a radial velocity of $1/3$ pixels per frame is shown in Figure 3. The object is moving in the direction of 340 degrees with respect to the horizontal left-to-right direction. Four different frames are given, i.e., the 1st, 64th, 128th, and 256th.

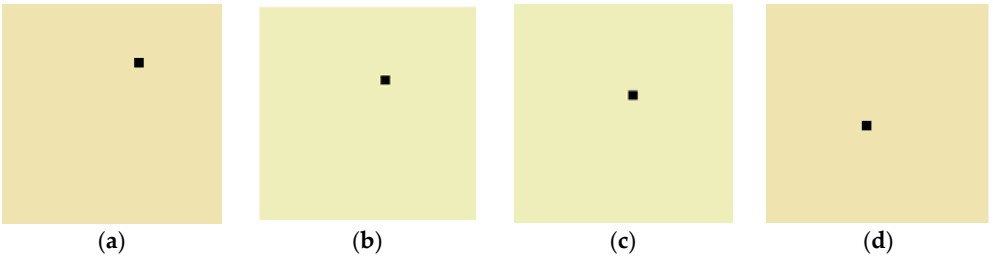

(a)       (b)       (c)       (d)

**Figure 3.** An object of size $10 \times 10$ pixels that is moving with a radial velocity of $1/3$ pixels per frame. The object is moving in the direction of 340 degrees with respect to the horizontal left-to-right direction. (**a**) Frame 1, (**b**) frame 64, (**c**) frame 128, and (**d**) frame 256.

The datasets used for experimentation covered a wide range of velocities with regard to the amplitude and direction. Specifically, six different radial velocities (amplitudes) were selected, i.e., $1/2$ (fast), $1/3$, $1/4$, $1/8$, $1/16$, and $1/32$ (slow) pixels per frame. For all these velocities, 24 different directions were chosen, with the first one at 0 degrees (horizontal direction from left to right) and counterclockwise every 15 degrees, as shown in Figure 4. Accordingly, a total of $6 \times 24 = 144$ different data cubes (velocities) were implemented.

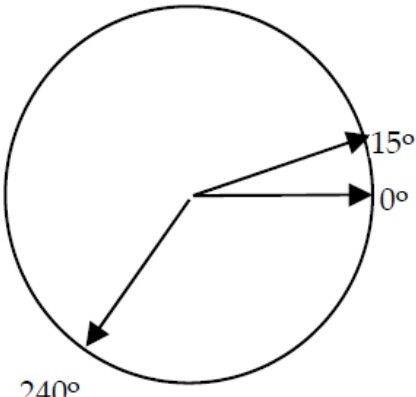

**Figure 4.** We chose 24 different directions in the experimental procedure, with the first one at 0 degrees (horizontal direction from left to right) and counterclockwise every 15 degrees.

Since the spectrum is a complex quantity, its amplitude and phase were evaluated separately. Parallel trajectories in the data cube that correspond to objects having the same velocity possess the same spectral amplitude information and differ in the phase information. Accordingly, regardless of the initial position of an object, its velocity corresponds to a specific amplitude of the spectral content. Thus, only the amplitude information is of interest and was recorded.

Studying the amplitude of the spectral content of a data cube, one can easily observe that from the total of $256^3$–$2^{24}$–16 million harmonics, only a small percentage has significant value. Thus, for each direction of the moving object with a specific velocity, a file that contained the positions of the most important harmonics (about 4000) was created, as those spectral components that are larger than the 12% (pixel value 30 with maximum 255) of the biggest spectral component are considered important harmonics.

However, later, when it is necessary to process complicated signals, the 20 largest harmonics among the 4000 will be recalled and their use in the final filter will be examined again. This is needed when in the signal to be processed, the vehicle to be recorded is not among the strongest objects.

To isolate an object moving with a specific velocity (velocity filtering) among other objects in a data cube, we need to perform the following steps:

1. Find the spectral content of the specific cube 3D FFT.

2. Eliminate from the spectral amplitude all harmonics except those corresponding to the specific velocity.

3. Evaluate the inverse 3D FFT to recover the data cube containing only the object with the specific velocity.

## 4. Implementation of Velocity Filtering Using QFT

### 4.1. Three-Dimensional Discreet Fourier Transform

The 3D discreet Fourier transform technique is a separable procedure. This comes from the fact that its expression

$$p(k_1, k_2, k_3) = \sum_{n_1=0}^{N-1} \sum_{n_2=0}^{N-1} \sum_{n_3=0}^{N-1} q(n_1, n_2, n_3) \, W_N^{k_1 n_1} W_N^{k_2 n_2} W_N^{k_3 n_3} \ 0 \le k_1, k_2, k_3 \le N-1 \tag{16}$$

can be written as follows:

$$p(k_1, k_2, k_3) = \sum_{n_1=0}^{N-1} W_N^{k_1 n_1} \sum_{n_2=0}^{N-1} W_N^{k_2 n_2} \sum_{n_3=0}^{N-1} q(n_1, n_2, n_3) \, W_N^{k_3 n_3} 0 \le k_1, k_2, k_3 \le N-1 \tag{17}$$

The first summation on the right of this equation means that we must perform $N^2$ N-point discreet Fourier transforms along the $n_3$ direction. Since each N-point FFT (for N being a power of 2) requires $N \log_2 N$ operations, for the implementation of the first summation of Equation (17), $N^3 \log_2 N$ operations are needed. The derived intermediate result is an $N^3$ complex cube to be again processed along the $n_2$ direction. Thus, another set of $N^3 \log_2 N$ complex operations is required. The second derived intermediate result is again an $N^3$ complex cube to be finally processed along the $n_1$ direction. Thus, a total of $3N^3 \log_2 N$ of complex operations are required. Following another way, we must perform N-point FFT along all horizontal lines, all horizontal columns, and all vertical columns regardless of the order in which they will be performed.

An example of 3D FFT is shown in Figure 5 for N = 4. From this figure and Equation (17), it is evident that the order of implementation of the 1D transforms, i.e., the order of the summations in Equation (17), does not matter. In image cubes (frame sequences), one of the dimensions is usually time.

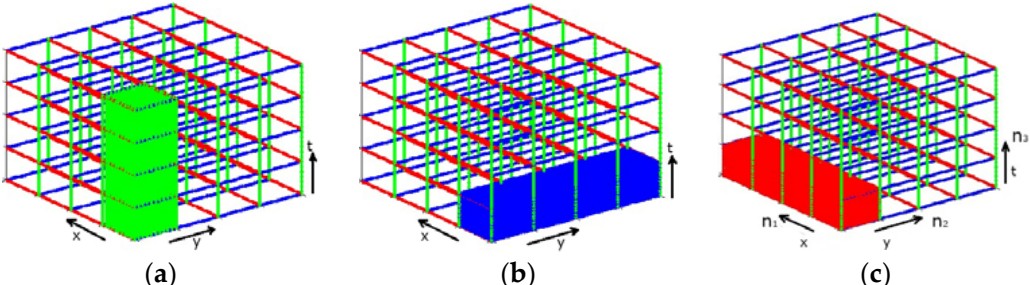

**Figure 5.** A 3D image cube of $4 \times 4 \times 4$ pixels. Since 3D FFT is a separable procedure, it can be performed in 3 phases (red, green, blue). In each phase, $4^2 = 16$ 1D FFTs are performed in directions (**a**) $n_3$ (green), (**b**) $n_2$ (blue), and (**c**) $n_1$ (red). In each phase, the result from the previous phase is used as input. The obtained result of the 3D FFT procedure is not related to the order in which the 3 above phases are applied.

*4.2. Quantum 3D Fast Fourier Transform*

A quantum cube can be represented using a quantum register Q constructed so that it encodes all required information, i.e., the position of a pixel in the frame (x, y), the serial number of the frame (time t), and the intensity or color (c) of the pixel [24]. We assumed that we have $2^k$ frames with $2^n \times 2^n$ pixels each and that the color of each pixel requires m bits for its color representation. In this case, a register $|P>$ having $2^n$ qubits is adequate for holding all position information, another register $|T>$ having k qubits will represent the time information, and a register $|C>$ with m qubits will represent $2^m$ different colors or grayscale levels. The register $|P>$ is separated into two sub-registers of n qubits each containing the row and column information in the form $|y>|x>$. The quantum register Q containing all the information about the quantum frame cube can be expressed as

$$Q = |C>_m \otimes |P>_{2n} \otimes |T>_k = \sum_{t=0}^{2^k-1} \sum_{i=0}^{2^{2n}-1} \sum_{j=0}^{2^m-1} a_{ijt} |j>|i>|t> \qquad (18)$$

In Equation (18), the coefficients $a_{ijt}$ for a specific t (frame of the cube) sum up to 1:

$$\sum_{j=0}^{2^m-1} |a_{ijt}|^2 = 1 \text{ for all i with } 0 \le i < 2^{2n} \text{ and all t with } 0 \le t < 2^k$$

and are used to express the color of a pixel with position *i* by means of a superposition of all possible colors. For a given pixel *i*, the coefficients $a_{ij}$ take the value of 1 if the color of the pixel is j and 0 otherwise. This is illustrated in Figure 6 with a simple example of a $2 \times 2 \times 2$ frame cube with eight colors (0–7). The corresponding coefficients $a_{ijt}$ for the specific examples in Figure 6 are as follows:

| *pct* | *pct* | *pct* | *pct* | *pct* | *pct* | *pct* | *pct* |
|---|---|---|---|---|---|---|---|
| $a_{000} = 1,$ | $a_{010} = 0,$ | $a_{020} = 0,$ | $a_{030} = 0,$ | $a_{040} = 0,$ | $a_{050} = 0,$ | $a_{060} = 0,$ | $a_{070} = 0$ |
| $a_{100} = 0,$ | $a_{110} = 0,$ | $a_{120} = 1,$ | $a_{130} = 0,$ | $a_{140} = 0,$ | $a_{150} = 0,$ | $a_{160} = 0,$ | $a_{170} = 0$ |
| $a_{200} = 0,$ | $a_{210} = 0,$ | $a_{220} = 0,$ | $a_{230} = 0,$ | $a_{240} = 1,$ | $a_{250} = 0,$ | $a_{260} = 0,$ | $a_{270} = 0$ |
| $a_{300} = 0,$ | $a_{310} = 0,$ | $a_{320} = 0,$ | $a_{330} = 0,$ | $a_{340} = 0,$ | $a_{350} = 0,$ | $a_{360} = 1,$ | $a_{370} = 0$ |
| $a_{001} = 0,$ | $a_{011} = 0,$ | $a_{021} = 0,$ | $a_{031} = 0,$ | $a_{041} = 0,$ | $a_{051} = 0,$ | $a_{061} = 0,$ | $a_{071} = 1$ |
| $a_{101} = 0,$ | $a_{111} = 0,$ | $a_{121} = 0,$ | $a_{131} = 0,$ | $a_{141} = 1,$ | $a_{151} = 0,$ | $a_{161} = 0,$ | $a_{171} = 0$ |
| $a_{201} = 0,$ | $a_{211} = 0,$ | $a_{221} = 0,$ | $a_{231} = 0,$ | $a_{241} = 0,$ | $a_{251} = 1,$ | $a_{261} = 0,$ | $a_{271} = 0$ |
| $a_{301} = 0,$ | $a_{311} = 0,$ | $a_{321} = 0,$ | $a_{331} = 0,$ | $a_{341} = 0,$ | $a_{351} = 0,$ | $a_{361} = 1,$ | $a_{371} = 0$ |

$$P = Position \quad c = Color \quad t = Time$$

In case we assume the time register $|T>$ as being the third dimension in our pixel cube representation without distinction between time and space (such data are available

in various cases, such as computed tomography), and furthermore, k equals n, then Equation (18) becomes

$$Q = |C>_m \otimes |P>_{3n} = \sum_{i=0}^{2^{3n}-1} \sum_{j=0}^{2^m-1} a_{ij}|j>|i> \tag{19}$$

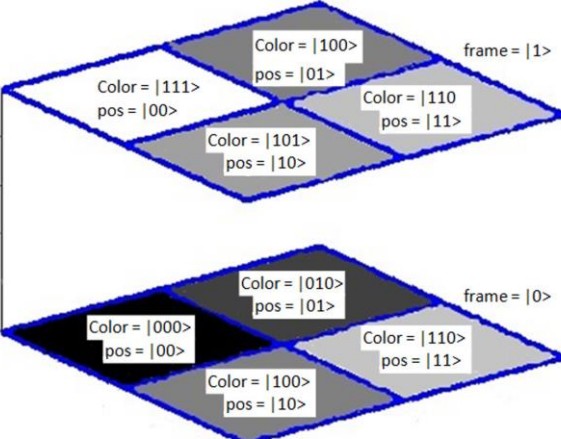

**Figure 6.** Example of a $2 \times 2 \times 2$ quantum cube sequence of frames with eight different colors. Three qubits are used to represent color information and the 3-pixel position information (in space and time).

In Equation (19), the coefficients $a_{ij}$ sum up to 1:

$$\sum_{j=0}^{2^m-1} |a_{ij}|^2 = 1 \ \forall i \ with \ 0 \leq i < 2^{3n} \tag{20}$$

and are used to express the color of a pixel with position $i$ by means of a superposition of all possible colors. For a given pixel $i$, the coefficients $a_{ij}$ take the value of 1 if the color of the pixel is j and 0 otherwise:

$$a_{00} = 1, \ a_{01} = 0, \ a_{02} = 0, \ a_{03} = 0, \ a_{04} = 0, \ a_{05} = 0, \ a_{06} = 0, \ a_{07} = 0$$
$$a_{10} = 0, \ a_{11} = 0, \ a_{12} = 1, \ a_{13} = 0, \ a_{14} = 0, \ a_{15} = 0, \ a_{16} = 0, \ a_{17} = 0$$
$$a_{20} = 0, \ a_{21} = 0, \ a_{22} = 1, \ a_{23} = 0, \ a_{24} = 1, \ a_{25} = 0, \ a_{26} = 0, \ a_{27} = 0$$
$$a_{30} = 0, \ a_{31} = 0, \ a_{32} = 0, \ a_{33} = 0, \ a_{34} = 0, \ a_{35} = 0, \ a_{36} = 1, \ a_{37} = 0$$
$$a_{40} = 0, \ a_{41} = 0, \ a_{42} = 0, \ a_{43} = 0, \ a_{44} = 0, \ a_{45} = 0, \ a_{46} = 0, \ a_{47} = 1$$
$$a_{50} = 0, \ a_{51} = 0, \ a_{52} = 0, \ a_{53} = 0, \ a_{54} = 1, \ a_{55} = 0, \ a_{56} = 0, \ a_{57} = 0$$
$$a_{60} = 0, \ a_{61} = 0, \ a_{62} = 0, \ a_{63} = 0, \ a_{64} = 0, \ a_{65} = 1, \ a_{66} = 0, \ a_{67} = 0$$
$$a_{70} = 0, \ a_{71} = 0, \ a_{72} = 0, \ a_{73} = 0, \ a_{74} = 0, \ a_{75} = 0, \ a_{76} = 1, \ a_{77} = 0$$

This is illustrated in Figure 7 with a simple example of a $2 \times 2 \times 2$ frame cube with eight colors (0–7).

The quantum register Q, expressed using Equation (18), was experimentally implemented by the circuit in Figure 8. It is in fact the quantum circuit that implements 3D velocity filtering.

As shown in Figure 8, at the output of the proposed quantum velocity filtering circuit, we find the input image cube. Additionally, exploiting the quantum interference phenomenon, we can use an additional qubit initially in state |0> to reinterpret the quantum image cube as a superposition of two image cubes. For example, for a high-pass or a low-pass 3D filter, the output image is the sum of the image cube containing high frequencies and the image cube containing the corresponding low frequencies. The additional qubit can be used to make the distinction between the two image cubes.

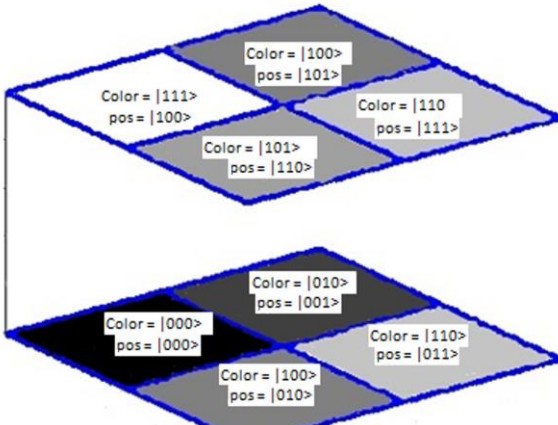

**Figure 7.** This is an example of a $2 \times 2 \times 2$ quantum cube with eight different colors. Three qubits are used to represent color information and 3-pixel position information. Such data are available in various cases, such as in computed tomography.

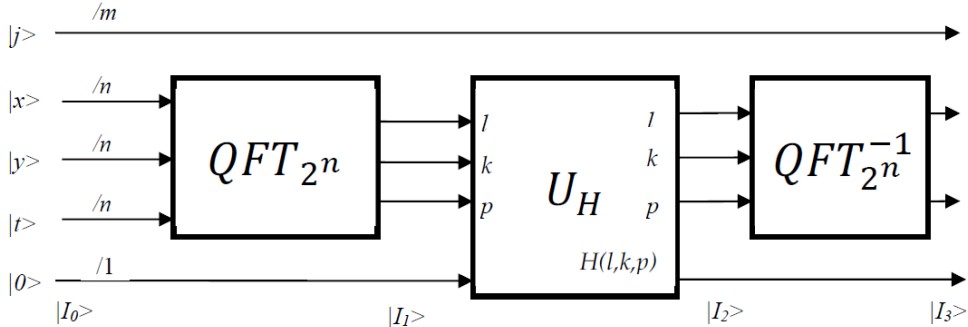

**Figure 8.** Implementation of the quantum circuit for 3D velocity filtering.

Next, we analyzed this process and described the state of the quantum velocity filtering circuit at each step of the computation, as marked in Figure 8. With respect to Equation (18), the register $|P>$ having $2n$ qubits and holding all position information is represented in the following Equation (21), with two registers of n qubits each corresponding to x and y positions, respectively. The input state $|I_0>$ is represented by the input image cube and an additional qubit in state $|0>$:

$$|I_0> \; = \; |Q> \; \otimes \; |0> \; = \; \frac{1}{2^n} \sum_{t=0}^{2^n-1} \sum_{y=0}^{2^n-1} \sum_{x=0}^{2^n-1} \sum_{j=0}^{2^m-1} a_{tyxj} |j> |t> |y> |x> |0> \tag{21}$$

where $|Q>$ holds the quantum image cube using the representation described previously. Applying 3D QFT on the image cube produces state $|I_1>$:

$$|I_1> \; = \; (I_m \otimes QFT_{2^{3n}})|Q> \otimes I|0>$$

$$= \frac{1}{2^n} \sum_{t=0}^{2^n-1} \sum_{y=0}^{2^n-1} \sum_{x=0}^{2^n-1} \sum_{j=0}^{2^m-1} a_{tyxj} |j> QFT_{2^{3n}}(|t> |y> |x>)|0>$$

$$= \frac{1}{2^n} \sum_{t=0}^{2^n-1} \sum_{y=0}^{2^n-1} \sum_{x=0}^{2^n-1} \sum_{j=0}^{2^m-1} a_{tyxj} |j> (QFT_{2^n}|t>)(QFT_{2^n}|y>)(QFT_{2^n}|x>)|0>$$

$$= \frac{1}{2^n} \sum_{t=0}^{2^n-1} \sum_{y=0}^{2^n-1} \sum_{x=0}^{2^n-1} \sum_{j=0}^{2^m-1} a_{tyxj} |j> \sum_{l=0}^{2^n-1} e^{\frac{2\pi itl}{2^n}} |l> \sum_{k=0}^{2^n-1} e^{\frac{2\pi iyk}{2^n}} |k$$
$$> \sum_{p=0}^{2^n-1} e^{\frac{2\pi ixp}{2^n}} |p> |0>$$

$$= \frac{1}{2^n} \sum_{t=0}^{2^n-1} \sum_{y=0}^{2^n-1} \sum_{x=0}^{2^n-1} \sum_{j=0}^{2^m-1} \sum_{l,k,p=0}^{2^n-1} a_{tyxj} e^{\frac{2\pi itl}{2^n}} e^{\frac{2\pi iyk}{2^n}} e^{\frac{2\pi ixp}{2^n}} |j> |l> |k> |p> |0> \qquad (22)$$

where $I$ and $I_m$ denote the identity operator on 1 and m qubits, respectively.

The next step performed by the quantum circuit is the equivalent of the classical filtering step. The state of the register holding the image cube does not in fact change to a state representing the filtered image cube, but rather, it undergoes an interference process with the additional qubit initially in state $|0>$. This is achieved using a quantum oracle built using the filter function $H(l, k, p)$.

The quantum state $|I_1>$ is a superposition of two states, a state representing the 3D frequencies that remain in the image cube and a state representing the 3D frequencies removed. Applying the oracle operator $U_H$ to this superposition, one can use the additional qubit to make the distinction between the two states. The oracle $U_H$ acts only on the position qubits and leaves the color qubits unaffected. Different choices of the filter $H(l, k, p)$ result in different selections of 3D velocity filters. So, the resulting $|I_2>$ can be the input to IQFT.

The last computational step in the quantum circuit in Figure 8 represents the IQFT that reverts from the frequency to the spatial representation of the image cube. The final state of the circuit contains the superposition of two quantum image cubes: the image cube containing the frequencies passed by the 3D filter and the image cube containing the frequencies suppressed by the 3D filter. The distinction between these two image cubes can be made using the additional qubit $|I_3>$. In fact, it can be interpreted as [1]

$$|I_3>=|Q_{unused\ velocities}> |0> + |Q_{used\ velocities}> |1> \qquad (23)$$

For extracting necessary information from the quantum-transformed image, a further processing step must be performed [1].

### 4.3. QFT Performance Versus FFT Performance

The computational performance of QFT is discussed in this subsection with regard to its superiority when compared to the computational performance of FFT. The required number of quantum gates is calculated, and a simple demonstration of their simplicity is presented with regard to the first simple necessary quantum gates. Note that the matrix in Equation (5) also implements classical FFT and performs the multiplication of the QFT matrix by the N × 1 column vector that contains the classical dataset. This multiplication would require $N^2$ operations. Therefore, we would expect classical discreet Fourier transform to require $O(N^2) = O(2^{2n})$ operations, which is exponential in n. The classical FFT algorithm can compute discreet Fourier transform in $O(N \log N)$ or $O(n2^n)$ operations, which is faster but still exponential in n.

However, QFT uses operation gates for its implementation. According to Figure 1, the number of operation gates used can be evaluated as follows:

1st row: 1 H gate + (n − 1)R gates = n gates
2nd row: 1 H gate + (n − 2)R gates = n − 1 gates
.
.
.

(n − 1)th row: 1 H gate + 1 R gate = 2 gates
n-th row: 1 H gate = 1 gate

Adding the gate count from each row gives n + (n − 1) + (n − 2) +⋯ + 1, or $O(n^2)$, gates, which is polynomial in n. Therefore, QFT is exponentially faster than discreet Fourier transform or FFT.

Furthermore, classical FFT (corresponding to 2 qubits, N = 4) in matrix form is

$$\vec{y} = U\vec{x} \tag{24}$$

where $U$ is

$$U = \frac{1}{2}\begin{pmatrix} 1 & 1 & 1 & 1 \\ 1 & i & i^2 & i^3 \\ 1 & i^2 & 1 & i^2 \\ 1 & i^3 & i^2 & i \end{pmatrix} \tag{25}$$

and we replace $exp(2\pi i/4) = i$. Equation (25) can be written as the product of two sparse matrices $U = U_1 U_2$, where

$$U_1 = \frac{1}{\sqrt{2}}\begin{pmatrix} 1 & 0 & 1 & 0 \\ 0 & 1 & 0 & 1 \\ 1 & 0 & i^2 & 0 \\ 0 & 1 & 0 & i^2 \end{pmatrix} \tag{26}$$

and

$$U_2 = \frac{1}{\sqrt{2}}\begin{pmatrix} 1 & 1 & 0 & 0 \\ 0 & 0 & 1 & i \\ 1 & i^2 & 0 & 0 \\ 0 & 0 & 1 & i^3 \end{pmatrix} \tag{27}$$

QFT is generated by matrix U and can be written as the product of four sparse matrices (the swap gate, the Hadamard gates (2 gates), and the controlled phase gate) as follows:

Swap gate

$$S = \begin{pmatrix} 1 & 0 & 0 & 0 \\ 0 & 0 & 1 & 0 \\ 0 & 1 & 0 & 0 \\ 0 & 0 & 0 & 1 \end{pmatrix} \tag{28}$$

Hadamard gate for the lower qubit

$$H_0 = \frac{1}{\sqrt{2}}\begin{pmatrix} 1 & 1 & 0 & 0 \\ 1 & -1 & 0 & 1 \\ 0 & 0 & 1 & 1 \\ 0 & 0 & 1 & -1 \end{pmatrix} \tag{29}$$

Hadamard gate for the upper qubit

$$H_1 = \frac{1}{\sqrt{2}}\begin{pmatrix} 1 & 0 & 1 & 0 \\ 0 & 1 & 0 & 1 \\ 1 & 0 & -1 & 0 \\ 0 & 1 & 0 & -1 \end{pmatrix} \tag{30}$$

Controlled phase gate

$$R_1 = \frac{1}{\sqrt{2}}\begin{pmatrix} 1 & 0 & 0 & 0 \\ 0 & 1 & 0 & 0 \\ 1 & 0 & 1 & 0 \\ 0 & 1 & 0 & i \end{pmatrix} \tag{31}$$

As shown, the sparse matrices of QFT are simpler than the classical matrices of FFT without dynamics. So, the calculations in QFT are simpler and faster. Similarly, in classical

FFT (3 qubits, n = 8), we have three sparse matrices, and in QFT, we have seven simpler sparse matrices, so the calculations are also faster.

Accordingly, for the N-qubit registers in QFT, a significant reduction is achieved in the required operations since all matrices involved in its evaluation are sparse. In the implementation of the example of Section 5.2, the execution time of conventional FFT was 200.63 s, while QFT implementation was carried out in 16.2 s.

## 5. Experimental Results

### 5.1. Moving Objects

In this subsection, the movement of some simple objects in a 3D scene is presented and their 3D spectrum is evaluated using the 3D QFT described by Equation (22). The data cube is considered to consist of 256 × 256 pixels and a total of 256 frames.

In Figure 9a, a small squared object in 3D space is shown. This thin (in the time axis—5 frames are occupied) object gives a horizontal spectrum with harmonics in a broad region as it is shown from various aspects of the spectrum (Figure 9b–d). Such an object, which would appear suddenly in the image cube for a small number of frames and then disappear, does not in fact exist. This means that the blue still object in Figure 9a is just a simulation to demonstrate its simple spectral content, and it exists only for five frames. Each direction in the spectral cube corresponds (inverse dimensions) to one of the three dimensions of the data cube, with a large extent of the one in Figure 9c, which is toward the vertical axis of the data cube (time). The three objects in Figure 9b–d in fact present the spectral content of the data cube in Figure 9a from different aspects.

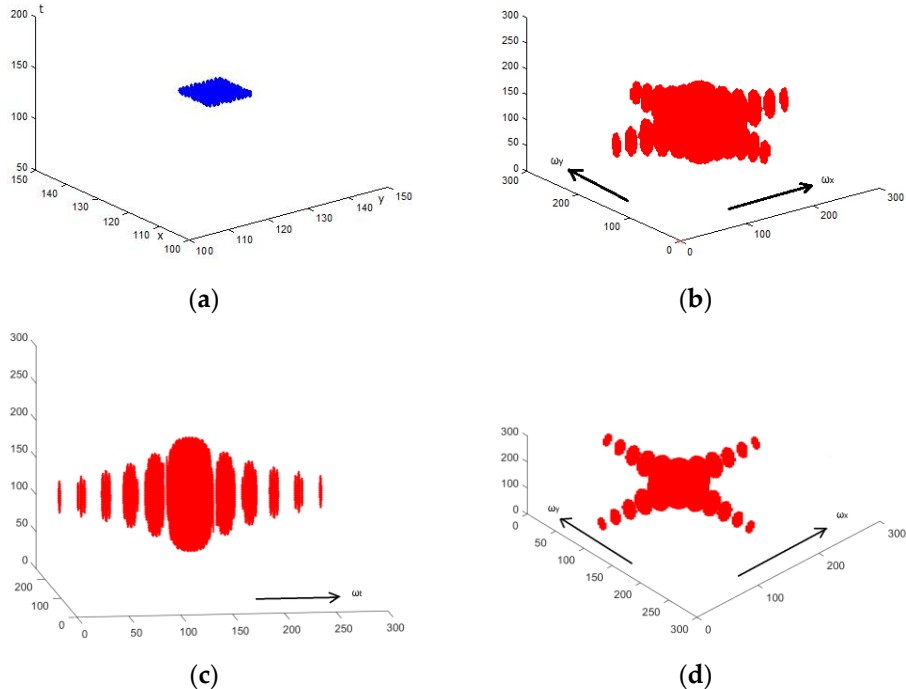

**Figure 9.** (**a**) A squared still object in the time domain (vertical axis). The object is present in the scene for just 5 frames. This kind of object seldom exists. (**b–d**) The object's 3D spectrum from various aspects. (**c**) The spectral direction (corresponding to $\omega_t$), which presents a large extent and corresponds to a small extent of the original data cube toward the vertical axis (t). (**b,d**) The extent of the spectrum toward the directions of the spatial frequencies from a different aspect (height) in the third axis $\omega_t$.

The object shown in Figure 10 cannot also exist in practice, since while unmovable, it appears in some of the frames and then disappears. However, its spectrum given in red becomes thinner compared to the previous one in Figure 9 and contains a smaller amount of spectral energy out of the horizontal spectral plane 129. In practice, a still object in space

should exist from 0 to 255 in the vertical axis. In this case, the spectrum is flat-lying only in the horizontal spectral plane 129 and corresponds only to frequencies irrelevant to the movement. So, the absence of movement corresponds to the spectral energy distributed in the horizontal plane 129 of 3D FFT.

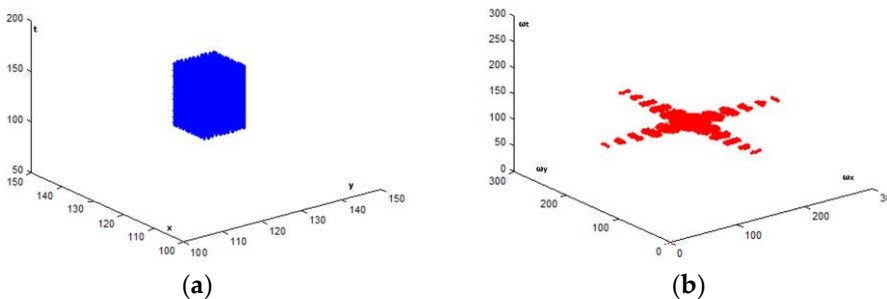

Figure 10. (**a**) A squared still object in the time domain (vertical axis). The object is present in the scene for 160 frames. (**b**) Its 3D spectrum is distributed mainly in plane 129 of the spectral cube.

In Figure 11, a moving object appears in the data cube in blue. The corresponding spectrum, in red, is flat and lies in a linear plane being vertical to the blue trajectory line. In this spectrum, some of the DC components exist in the horizontal spectral plane 129. This is a problem when we try to get rid of the (still) background. In Figure 12, a similar case is examined, where the object is moving in a different direction and its spectrum (red) is again oriented perpendicular to the blue trajectory.

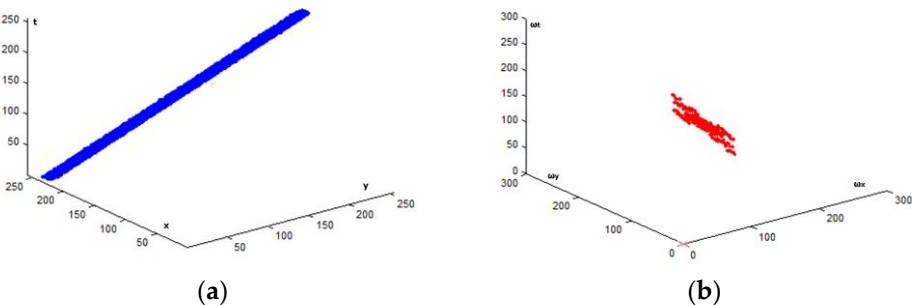

**Figure 11.** The evaluation of the spectra of a moving object. (**a**) The trajectory of the moving object in the data cube (blue) and (**b**) the 3D spectrum of this trajectory is flat (red). The direction of the trajectory can be considered as being perpendicular to the plane of its spectrum.

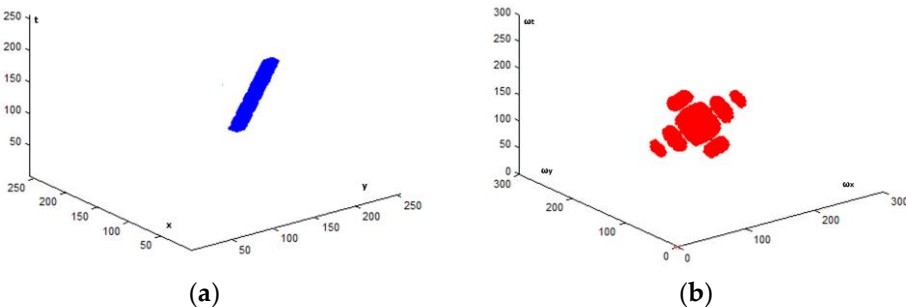

**Figure 12.** A different case of the 3D spectrum of a moving object. (**a**) A moving object (blue) in a different direction than that in Figure 11. (**b**) Its flat spectrum in red. Again, the trajectory of the object is perpendicular to the plane of its spectrum.

Finally, in Figure 13, both moving objects exist in the data cube, and their (red) spectrum consists of the combination of the spectra in Figures 11 and 12. Evidently, when one of the two objects must be rejected (filtered), the common frequency components need special care.

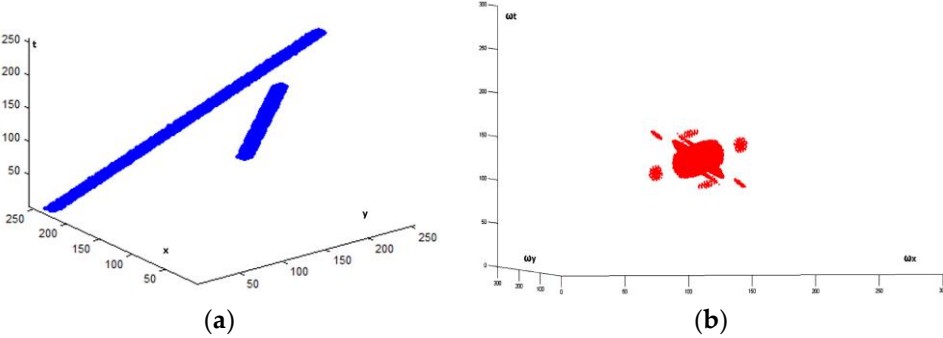

**Figure 13.** The case of two moving objects in the data cube and their 3D composite spectrum. (**a**) The two moving objects (blue) with different trajectories. (**b**) Their spectra (red) with planes being perpendicular to the corresponding trajectories.

According to the described capabilities of the network in Figure 8, we need to evaluate the 3D spectrum of a specific trajectory and then devise the circuit $U_H(l, k, p)$ so that only specific frequencies are selected according to the 3D filter bank with various versions of $U_H(l, k, p)$ that have been built. This is in fact the procedure described in Section 3.

The selection of the filter $U_H(l, k, p)$ is initially made for a specific filtering transfer function according to the trajectory that is necessary to be isolated. The block diagram in Figure 8 contains a unique filtering capability and, of course, its complement. To isolate a different trajectory, another block diagram should be activated.

### 5.2. QFT Velocity Filtering Example

A bank of filters $U_H(l, k, p)$ was devised in an analogous manner to that in [28]. This filter bank was used to isolate cars moving with a specific velocity on a bridge. In Figure 14, two frames of a video from the bridge are shown, the 360th and the 380th. The white car moving from left to right at the bottom of the scene was isolated. All the frames were converted from color to grayscale using the MATLAB rgb2gray utility. Simultaneously, the length of the scenes was restricted to 512 pixels, and 512 frames were used so that a data cube of $512 \times 512 \times 512$ was processed. Frames 360 and 380 in grayscale and restricted in length to 512 pixels are shown in Figure 15. After the appropriate selection of the filter $U_H(l, k, p)$, inverse 3D QFT was applied, and the obtained result is shown in Figure 16. The obtained 360th and 380th frames contained only the white car that was moving at the bottom of the scene from left to right. This was possible since the spectrum of the trajectory of this car was recorded in one of the filters in the filter bank and was appropriately loaded in the block $U_H(l, k, p)$.

In Figure 17, the selection of the filter $U_H(l, k, p)$ used in this experiment is demonstrated, providing three frames containing the specific car isolated from Figure 16 (Figure 17a–c), along with a sketch in the 3D data cube of the trajectory of this car (Figure 17d) and its corresponding spectral content $U_H(l, k, p)$ (Figure 17e).

As stated in the Introduction section, the isolation of the trajectory of a moving object within a specific range of speeds provides the capability of exactly evaluating the spectral content of the specific trajectory. The quantum oracle used incorporates this spectral content in a special way. Specifically, the |l>, |k>, and |p> registers in Figure 8 contain the whole spectrum of the image cube, but only the spectral content of the specific trajectory is left to pass through $U_H$ (filtering). Accordingly, the quantum oracle $U_H$ reorganizes the contents of the |l>, |k>, and |p> registers so that only the spectrum of the trajectory of the specific moving object remains in the data cube. The output of $U_H$ is shown in Figure 17e. Furthermore, based on the quantum interference phenomenon, we can use an additional qubit initially in state |0> to reinterpret the quantum image cube as a superposition of two image cubes, namely the spectral content of the trajectory (Figure 17e) and its complement.

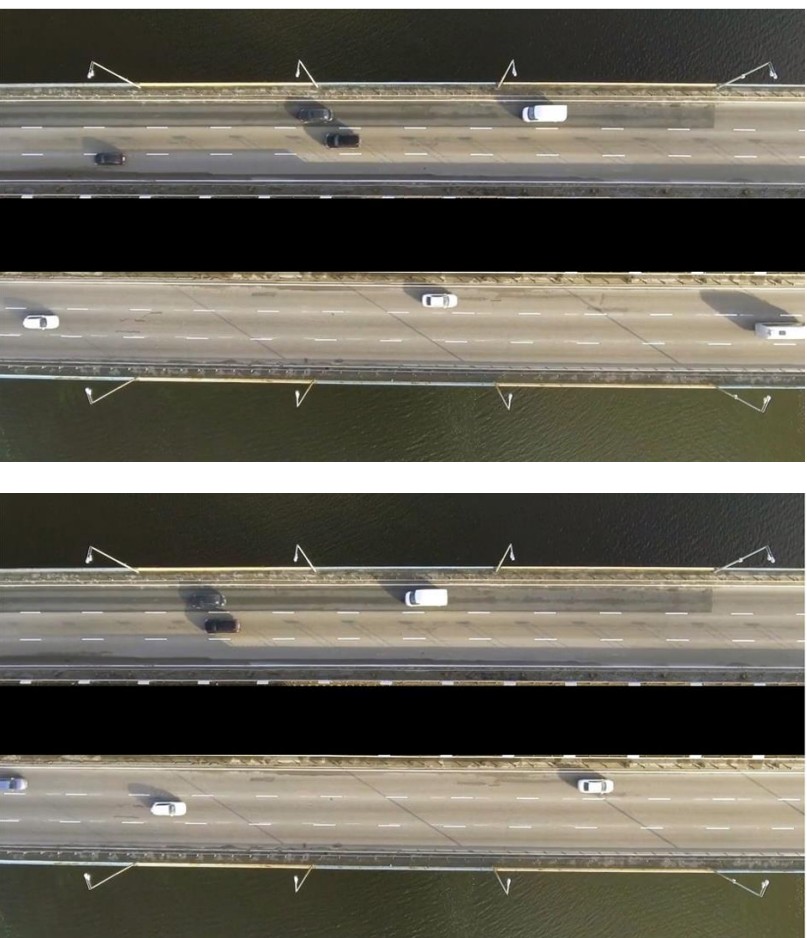

**Figure 14.** The video of the bridge. Scenes 360 (**upper**) and 380 (**lower**). The velocity of the white car at the bottom of the two scenes was recorded, and this car was isolated using the 3D QFT algorithm.

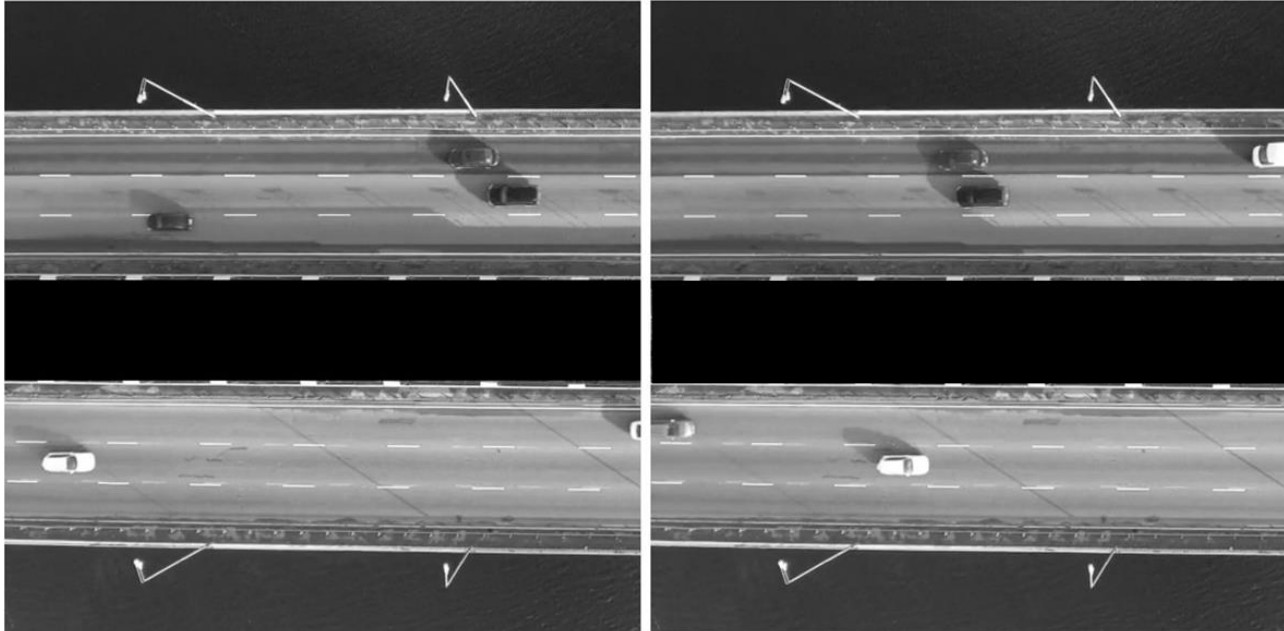

**Figure 15.** The video of the bridge was converted to grayscale, and the total data cube was $512 \times 512$ pixels $\times$ 512 frames. Frames 360 (**left**) and 380 (**right**) are shown.

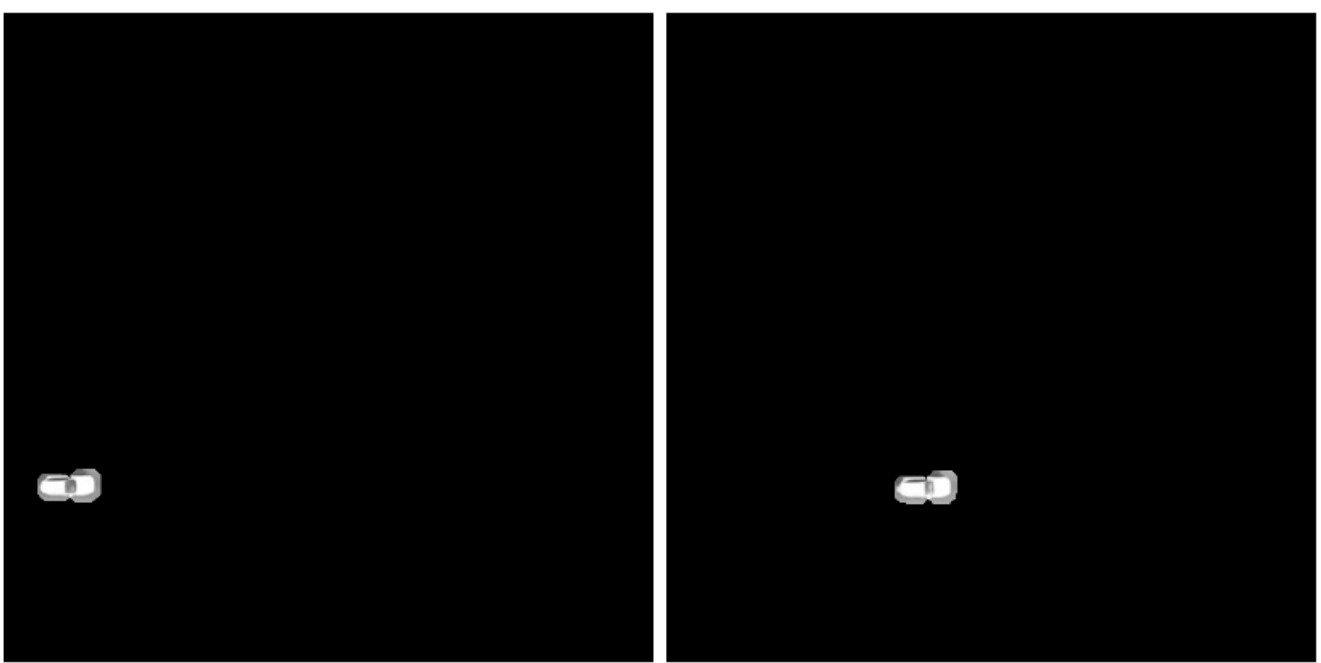

**Figure 16.** After 3D QFT processing and application of inverse 3D QFT, frames 360 (**left**) and 380 (**right**) contained only the car with the specific velocity (velocity filtering).

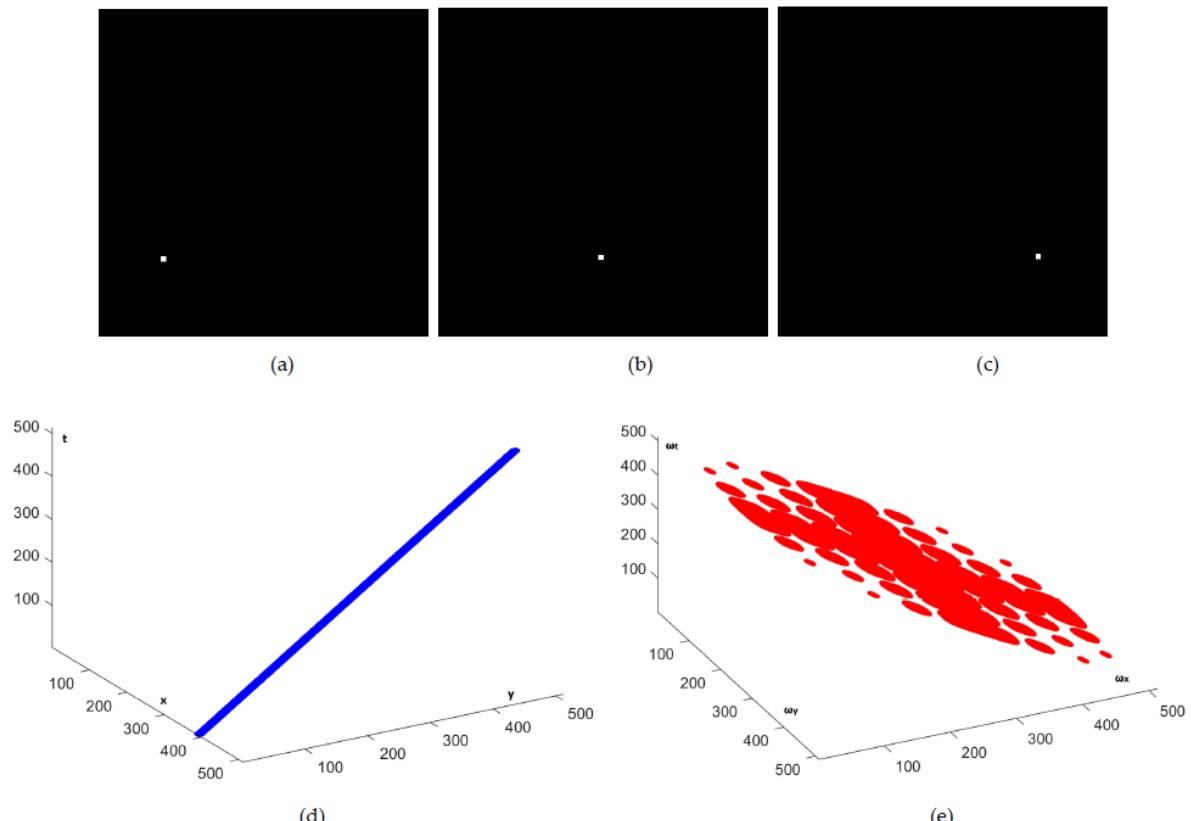

**Figure 17.** The selection of the filter $U_H(l, k, p)$ capable of isolating the specific car from Figure 16. Three different frames with the isolated car, the 100th (**a**), the 250th (**b**), and the 400th (**c**), along with a sketch in the 3D data cube of the trajectory of this car (**d**), and its corresponding spectral content $U_H(l, k, p)$ (**e**).

Experimentally, we ran quantum simulations on a conventional computer running MATLAB. All procedures were based on the approach the authors in [36] specified. Comparisons with conventional approaches proved that the execution time for QFT is smaller in the order of 100 when a quantum circuit is used. This significant reduction is achieved since all matrices involved in its evaluation are sparse. In the implementation of the simulation experiment, the execution time of conventional FFT was 200.63 s, while QFT implementation was carried out in 16.2 s.

## 6. Conclusions

The quantum version of 3D FFT was presented analytically in this work, emphasizing the detailed explanation of its application to a specific example of a data cube coming from an ordinary video. The form of the development of 3D QFT has an important tutorial character. Simultaneously, it constitutes an important technical utility for isolating objects that are moving at speeds within certain limits. This utility can be used for numerous applications.

A filter bank was built having a series of quantum filter functions $U_H(l, k, p)$ that differ in their filtering capabilities with regard to the velocity of the object as well as its direction. The filter function is selected based on the properties of the object that is to be isolated.

The performance of the quantum circuit in Figure 8 is effective since the object with the specific velocity (white car at the bottom of the scene moving from left to right in Figures 14–16) is totally isolated as the spectral signature of its trajectory (Figure 17e) is embodied as the operational content of the quantum oracle $U_H$. Simultaneously, other cars with the same velocity are also isolated and brought to the foreground in the final scene.

Specific quantum velocity filtering techniques were not found in the literature for comparisons to be carried out. However, our approach was based on [28] only with regard to the organization of the filter bank. Its superiority in performing fast calculations for isolating objects with a specific velocity is clear.

**Author Contributions:** Conceptualization, G.K. and V.A.; realization, G.K. and V.A.; presentation, G.K. and V.A. All authors have read and agreed to the published version of the manuscript.

**Funding:** This research received no external funding.

**Institutional Review Board Statement:** Not applicable.

**Informed Consent Statement:** Not applicable.

**Data Availability Statement:** Not applicable.

**Conflicts of Interest:** The authors declare no conflict of interest.

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
