# Peer review of "Velocity Filtering Using Quantum 3D FFT"

_photonics, doi:10.3390/photonics10050483_

Round 1

Reviewer 1 Report

The velocity filters are implemented by means of the quantum 3D FFT. This approach results in velocity filtering in comparatively short time. The necessary quantum computational units to implement the quantum 3D FFT are reviewed, and the application examples of velocity filtering are given.

The motivation is not clear.

Quantum transforms and quantum algorithms are very popular topics in the field of quantum information and quantum computing. Some references are piled up and should be described seperately. And some new progress should be added and introduced.Signal Processing: Image Communication 110 (2023): 116891. Physica A: Statistical Mechanics and its Applications 605 (2022): 128017. 

The sentences are complicated and the expression is too poor, for example, are build, one can extend to N-qubit QFT.

The Dirac symbols are not correct.

The results should be explained in detail, especially from the theoretical aspect.

More comparisons would be better to convince the audience.

It is original and interesting somewhat, I think it can be accepted after minor revision.

Reviewer 2 Report

The authors present an implementation of velocity filtering using a quantum 3D-FFT approach. An example which illustrates the utility of the implementation in a scenario for isolating moving cars is also given. The authors should take into account the following comments and suggestions in order to improve the overall quality of the paper.

-          The contributions of the paper should be described in more detail. The single paragraph that mentions the contributions in section 1 should be extended.

-          The own previous work of the authors, although cited (references 26-27) should be specifically mentioned, in order to highlight the novel contributions of the current work.

-          Title of section 4 should be Implementation of Velocity Filtering using QFT.

-          …is presented… instead of …are presented… (row 400).

-          Figure 9 is not clearly explained. What exactly are the various aspects to which subfigures b, c, d are referring to? Some measurement units would also be necessary on the three axes.

-          …examined. instead of …examine. (row 428).

-          …one… instead of …the one… (row 432).

-          The phrase from rows 445-448 is not grammatically correct and should be rephrased.

Reviewer 3 Report

see attached

Round 2

Reviewer 3 Report

see attached

Round 3

Reviewer 3 Report

see attached
